# Awareness and Performance towards Proper Use of Disinfectants to Prevent COVID-19: The Case of Iran

**DOI:** 10.3390/ijerph18042099

**Published:** 2021-02-21

**Authors:** Zahra Safari, Reza Fouladi-Fard, Razieh Vahidmoghadam, Mohammad Raza Hosseini, Abolfazl Mohammadbeigi, Alireza Omidi Oskouei, Mostafa Rezaali, Margherita Ferrante, Maria Fiore

**Affiliations:** 1Research Center for Environmental Pollutants, Department of Environmental Health Engineering, Faculty of Health, Qom University of Medical Sciences, Qom 3715614566, Iran; zsafari353@gmail.com (Z.S.); r1375vahid@gmail.com (R.V.); 2Student Research Committee, Qom University of Medical Sciences, Qom 3715614566, Iran; m.reza68.h@gmail.com; 3Department of Public Health, Faculty of Health, Qom University of Medical Sciences, Qom 3715614566, Iran; beigi60@gmail.com (A.M.); omidioskouei@gmail.com (A.O.O.); 4Independent Researcher, Isfahan 8157983853, Iran; mostafarezaali@gmail.com; 5Department of Medical, Surgical and Advanced Technologies “G.F. Ingrassia”, University of Catania, 87-95123 Catania, Italy; mfiore@unict.it

**Keywords:** awareness, practice, disinfectant, COVID-19, SARS-CoV-2, geographical distribution

## Abstract

This study aimed to assess the awareness and performance of Qom citizens towards using disinfectants and compared its relationship with geographical distribution of COVID-19 outbreak in Qom, Iran. The study was conducted by a researcher-made questionnaire during April and May, 2020. COVID-19 incidence data for each district of city was obtained from health department of Qom province. Data were analyzed using Excel, SPSS and ArcView (GIS) softwares. It was found that the highest level of citizens’ awareness (52%) was in the weak range while their performance (56%) was in the good range. According to Spearman’s correlation analysis, there was a strong correlation (rho 0.95) between the total mean of awareness and performance (*p* < 0.01). The highest incidence rate of COVID-19 was in district 7 which had the lowest mean score in both awareness and performance. In addition, the results of ANOVA (LSD—least significant difference) showed that there was a significant difference (*p* < 0.05) between district 7—with lower mean scores in awareness and performance—and other districts. Overall, it is concluded that citizens’ awareness level was lower than that of their performance. This conclusion not only calls for more training programs to be implemented in public places, schools, universities and governmental offices, but it also necessitates maintaining a proper and timely training about using disinfectants.

## 1. Introduction

In the late December 2019, the outbreak of the SARS-CoV-2 virus was reported in Wuhan, China [1]. The disease was identified in the Huanan Seafood Market. The World Health Organization (WHO) described it as a public health and global concern on January 30th and a pandemic on March 11th [2]. Coronaviruses are a large family of viruses that can cause a variety of illnesses, ranging from the common cold to acute and severe respiratory syndrome [3] which is common in both humans and animals [4]. Since this is a newly identified virus, new reports about different facets of COVID-19 are being released almost every day during the pandemic. However, the common symptoms include fever, respiratory symptoms, cough, fatigue, and breathing difficulties [5]. The crude mortality ratio for COVID-19 is between 3–4% [6] while in another study the mortality rate has been reported to be 2.9% [7]. As of 2:45 P.M. 6 September 2020, 2026 new cases, 382,772 confirmed cases, and 22,044 deaths from COVID-19 were reported in Iran [8]. In addition to mortality, COVID-19 pandemic has had the vastest impact on human health on a global scale [9], which has caused limitations and consequences in medical, social and economic systems. Its implications include safety of health care providers, economic safety, food safety, attention to mental health and the need to implement a wide range of crisis management measures [10].

The risk of respiratory failure in patients infected with SARS-CoV-2 is serious and requires urgent care support [11]. Due to similarities between the SARS-CoV-2 virus and SARS and despite the increasing public awareness, SARS-CoV-2 transmission is becoming more widespread than SARS [12]. The virus is transmitted from person to person through droplets and contact with people or objects [13]. As the virus continues to spread, member states of WHO are considering and releasing guidelines [14] to prevent the further spread of disease to new districts or reduce human-to-human transmission in districts where the SARS-CoV-2 virus is currently spreading. Public health measures to achieve these goals may include quarantine, such as restricting movement or separation of healthy people from the rest of the population, with the aim of monitoring the symptoms and early detection [15]. There is also a possibility of transmission of SARS-CoV-2 during recovery period, which makes controlling the outbreak more arduous [11].

The most common and important strategy for dealing with a communicable disease (like COVID-19) is a public health approach [16]. According to recommendations of WHO, frequently touched surfaces such as bedside tables, bedframes, bedclothes, and other bedroom furniture, bathrooms and toilet surfaces and devices such as clothes, baths and hand towels used by a patient, a healthy person or suspected patients should be disinfected daily with household disinfectants. This is especially crucial for people who take care of quarantined patients at home [15]. Recent studies have proved that SARS-CoV-2 can last from several hours to several days on different surfaces, for instance 4–5 days on wood, and less than 8 h on latex gloves [4]. WHO also provides guidance on individual equipment for infection prevention and control, which includes using eye protection (glasses), facemasks, and avoiding touching mucous membranes (eyes, nose, or mouth). Thorough hand washing is the usual measure to prevent transmission of infection, which must be done carefully. These health standards should be taken into account by health care staff and the general public [17].

Nowadays hospitals are mainly using chemicals as disinfectants. Some chemicals such as hydrogen peroxide, formaldehyde, and glutaraldehyde are used as disinfectants to inactivate viruses such as SARS-CoV-2 [18]. To ensure environmental disinfection, WHO recommends that disinfection procedures be performed appropriately and consistently. Most high-touch surfaces such as doors, toilets, tables, switches, etc., should be regularly disinfected with household disinfectants [4]. Studies have also shown that upon the arrival of spring and summer and with the rise in temperature and humidity, the number of cases will not be reduced unless people comply fully with the health standards. Therefore, it is indispensable to observe hygienic standards which can play a significant role in reducing the transmission of the virus [19]. Accordingly, people’s preparation and training to deal with this contagious disease and subsequently to control it will be very productive in this regard.

Some countries, including Iran, have witnessed a rapid spread of this disease [20,21]. The city of Qom in Iran was one of the first cities to be infected with this virus. To put it more accurately, in Iran, the first confirmed case of COVID-19 was reported in Qom. According to recommendations of WHO, health care (including proper and timely use of disinfectants) is one of the basic principles of virus prevention and control. Therefore, due to the need to pay due attention to health care during COVID-19 pandemic, this study set out to assess the awareness and performance of Qom citizens towards using disinfectants and compare its relationship with geographical distribution of COVID-19 outbreak in Qom.

## 2. Materials and Methods

The present study was approved by Ethics Committee of Qom University of Medical Sciences (ID: IR.MUQ.REC.1399.010).

This study is a descriptive one using a researcher-made questionnaire as its data-collection tool, which was conducted among people living in the city of Qom, Iran during April and May (refer to the Appendix A for the questionnaire). This city with eight urban districts (as shown in Figure 1), is the capital of Qom province and located 130 km to the southwest of Tehran. Located in the central district of Iran, it has a dry and semi-dry climate with an annual rainfall of 161 mm [22,23,24,25,26].

The researchers designed the questionnaire dealing with the methods of preparing and using disinfectants in the COVID-19 crisis based on information obtained from WHO [27,28]. The questionnaire was designed in the form of 40 items and three sections including personal and demographic information, awareness, and performance. The questionnaire, then, underwent pretesting. The validity and reliability of the questionnaire were confirmed by a panel of experts consisting of faculty members of the School of Health, Qom University of Medical Sciences. The content validity ratio (CVR) and content validity index (CVI) were found to be 0.878 and 0.775, respectively. To select the samples from different parts of the city as well as to reduce the risk of being exposed to COVID-19, it was decided that the questionnaires be designed and completed online.

The personal and demographic sections of the questionnaire included seven items about age, sex, level of education, marital status, occupation, residential district, and monthly income. The awareness and performance section also had 10 and 18 items, respectively. The items in this section were designed based on health protocols provided by WHO, which included questions about ordinary people’s awareness of how to use disinfectants, how to make them, etc. For instance, one of the questions was “Which alcohol is used as a disinfectant?” The correct answer is “ethanol” out of the four options—ethanol, methanol, both, do not know. For the following item, “the most effective concentration of alcohol for disinfection”, “70%” is the correct answer among the options. The performance section was also designed based on health protocols provided by WHO, consisting of five Likert-type options of “always, most of the time, sometimes, rarely and never”. For instance, some of the questions were as follows: “Do you wash your hands when you get home? Do you disinfect your hands or use gloves while buying items such as bread? Do you use special disinfectants to disinfect fruit and vegetables?” This section also included items about people’s health performance regarding proper and timely use of antiseptic agents before COVID-19 crisis, like “Did you use to do it (before the COVID-19 pandemic)?” followed by “yes” and “no” options. Questions 36 to 40 dealt with whether people receive health information through media such as WhatsApp, Telegram, Instagram, television, or radio.

These items were entered onto the “Porsline” website and then sent to citizens in form of messages in WhatsApp and Telegram applications. To observe ethical considerations, the purpose of this study was embedded within the sent messages and thus the citizens were informed of the purposes of the study. They were also asked to answer questions with sufficient care and attention.

Since an appropriate number of questionnaires were to be completed based on the population of each municipality district, it was decided that in addition to the use of cyberspace, several questionnaires be completed in person. Therefore, according to the population of each municipality district and the number of required questionnaires, several hard copies of the questionnaires were prepared and distributed among local business people who were residents of that municipality districts.

The sample size was calculated based on Cochran’s formula. The sample size in each city was calculated taking into account the household population and according to the following formula:(1)$$n=\frac{\frac{Z^{2}\mathrm{pq}}{d^{2}}}{1+\frac{1}{N}\left(\frac{Z^{2}\mathrm{pq}}{d^{2}}-1\right)}$$
where *N* (statistical population—number of households in Qom) was 99,572 households according to the latest official statistics of the country, assuming d = 0.05, z = 1.96, *p* and q = 0.5, the sample size was calculated to be 384.

COVID-19 incidence data for each urban district were obtained from the health department of Qom province. Finally, the data were analyzed using Excel (Microsoft, Redmond, WA, America), SPSS (IBM, Chicago, IL, America), and Arcview (GIS) softwares (ESRI, Redlands, San Bernardino, CA, America). Total awareness and practice scores were calculated according to the following scale:

**Total awareness**: The scores below 4 = weak, between 5 and 7 = average, above 8 = good.

**Total practice**: The scores below 20 = weak, between 40 and 21 = average, above 41 = good.

Independent Samples *t*-Test was used to analyze the differences among group means with regard to sex and marital status. One-way analysis of variance (ANOVA) was used to analyze the differences between group means in the ordinal and categorical variables (e.g., education level). Responses to items between pre- and post- COVID-19 outbreak were compared using the McNemar test. To analyze these questions by McNemar test, the ordinal and categorical variables were converted to nominal variables. Finally, the correlation between quantitative variables was analyzed by Spearman’s correlation. All data analysis was carried out at a significance level of less than 0.05.

## 3. Results

### 3.1. Socio-Demographic Information

A total of 412 questionnaires were completed online, 89 of which were excluded due to the absence of necessary information and distortion of information (including those respondents under the age of 20). Then, because of the lack of sufficient completed questionnaires (in this study, the entire eight districts of the municipality were considered in distributing the questionnaires), 77 questionnaires were completed in person in compliance with health protocols. Finally, a total of 400 final questionnaires were included in the study.

As shown in Table 1, the majority of the respondents were female (61.5%), married (83.25%), and in the age range of 31–40 (42.75%), while the age group of over 60 years comprised the lowest number of the respondents (0.75%). It is worth mentioning that some of datasets had missing values.

### 3.2. Awareness

Table 2 shows the level of awareness of respondents. According to this table, the highest level of awareness of respondents was reported for the question “How many dishwashing liquid drops are needed for pre-disinfection of fruits” (71.25% of the respondents answered this question correctly). Furthermore, the respondents identified methanol as toxic, industrial alcohol, and ethanol 70% as the most appropriate medical disinfectant alcohol. On one hand, 55% of respondents had no information about the percentage of chlorine in bleach liquids. While on the other hand, only 15.25% of respondents knew that chlorine, present in bleach, is in the form of a solution with a 5% concentration. It should be mentioned that all datasets had missing values.

### 3.3. Performance

Appendix A shows the distribution of responses relating to performance. The results show that 88% of respondents washed their hands immediately upon arrival at their houses after the outbreak, while 61% of them used to do so before the outbreak. Although most of the respondents were successful in the proper implementation of hygienic protocols such as hand washing when buying bread (54%), avoiding touching eyes and face with disinfected hands (59.75%), and proper hand washing for 20 s (63.25%), they were not used to carrying out these measures before the outbreak. The analysis of questionnaires demonstrated that 23.25% of citizens did not follow the necessary protocols for disinfecting fruit and vegetables with special disinfectants after the outbreak. On the other hand, the lack of implementation of these protocols before the outbreak had the largest share (57.5%). Furthermore, McNemar test revealed that there was a significant difference (*p* = 0.00) between pre- and post-COVID-19 outbreak for questions (11 vs. 12), (14 vs. 15), (16 vs. 17), (19 vs. 20), (26 vs. 27). In addition, it was found that the citizens followed the news and information about COVID-19 disease more through television (79.75%) and had less trust in cyberspace. However, the use of the Telegram application was more than those of others. However, it should be mentioned that all datasets had missing values.

### 3.4. Total Awareness and Performance Scores

According to Table 3, the total mean scores of awareness (out of 10 points) and performance (out of 57 points) were calculated to be 4.4 ± 2.2 and 40.2 ± 7.8, respectively. Independent *t*-test indicated that there was no significant difference between awareness and performance among men and women, but women had a higher mean score in awareness (4.7 ± 2.1) and performance (40.6 ± 7.6) than men. As to the municipality districts, the highest mean score of awareness was related to municipality district 1 (5.1 ± 2.4) while the highest mean score of performance was related to municipality district 4 (43.0 ± 7.1).

In addition, the lowest mean score for both awareness (2.6 ± 1.8) and performance (36.6 ± 6.9) belonged to municipality district 7 (Figure 2).

Moreover, one-way analysis of variance (ANOVA) showed that not only there was a significant difference between total means of awareness and variables of age, district, level of education, and job but also a significant difference was observed between the total means of performance and variables of the district, level of education and job (*p* < 0.05). Subsequently, according to Spearman’s correlation analysis, there was a strong correlation between the total mean of awareness and performance (rho 0.95, *p* < 0.01).

Figure 3 shows the citizens’ level of awareness and performance as classified into three levels of weak, middle, and good. Accordingly, it was found that most citizens’ total awareness (a) (52%) was in the weak range (below 4) and their performance (b) (56%) was in the good range (above 41).

### 3.5. Relationship between Mean Scores of Awareness and Performance and COVID-19 Incidence

The incidence was calculated by the following formula:(Frequency of disease/population) ×10,000.(2)

As shown in Figure 4, the results showed that the highest rate of incidence of COVID-19 was in municipality district 7 (15.4 cases per 10,000 population), while the lowest was in municipality district 8 (3.6 cases per 10,000 population). The sequence of this event was established between the municipality district in the following order: 7 > 3 > 1 > 5 > 4 and 6 > 2 > 8 (Figure 4 and Table 4). On the other hand, as shown in Table 5, the results of ANOVA (LSD) revealed that compared with other districts, a significant difference was observed in municipality district 7 (*p* < 0.05) with lower mean scores in awareness and performance.

## 4. Discussion

As the most common and essential strategy for this current situation is via the public health approach [16], this study was an endeavor to shed light on the awareness and performance of Qom citizens in using disinfectants. However, it should be mentioned that, thus far, no study has been conducted for this purpose in Qom. Since women constituted the majority of participants as in other similar studies and subsequently had higher awareness and performance scores than men [29,30,31], it might be concluded that women care more about their own and family health. It is worth saying that we did not impose any limitations other than that pertaining to “over the age of 20”. However, due to the fact that most of the questionnaires were completed online, there were more female participants than men, so our study was inadvertently biased and women constituted the majority of participants. Not surprisingly, most of the participants in the study were housewives. This result is in contrast with the results of a study carried out in Pakistan in which it was shown that men had a better attitude towards the prevention of infection than women [30]. The current study showed that mass media (e.g., radio, TV, etc.) still have a greater and essential impact on people’s awareness and attitude than virtual networking applications [32]. This finding contrasts with the findings of a study in Pakistan in which most participants would get their information from social media [30].

Knowing how to raise knowledge and awareness of people is a concern for the government. Since people would prefer television to Telegram as their main source of obtaining information, it goes without saying that information about disinfectants like the methods of diluting them, their specific differences, the uses of each one, etc., should be broadcast in these channels. As some people may have access to only one of the medias, it might be suggested that training and developing awareness and performance should be based on proper training in both mass and virtual media.

As mentioned in the results section, citizens’ awareness level was lower than that of their performance. One of the reasons for this might be continuous mass media training on how to use disinfectants and their emphasis on proper performance. On the other hand, mass media have played a greater role in educating people, but this awareness is not basic and scientific but rather functional. In a study conducted by Taghrir.et al on medical students, it was found that a great deal of participants’ awareness and performance was at a high level, which may be due to their job and profession [29]. According to the results of the current study, there was a strong correlation (with rho 0.95) between the total mean of awareness and the total mean of performance in each municipality district, hence those districts with higher awareness level tended to perform better.

In this study, the total awareness of people was lower than their total performance. Considering the high correlation between awareness and performance and the finding that most people obtain their information from mass media, it is, therefore, suggested that mass media should focus on increasing scientific and basic awareness in addition to functional one. Furthermore, based on the results, municipality districts 4 and 1 have higher levels of education, culture, and income than other districts while municipality district 7 mostly contains immigrants and travelers. It might be contributed to the fact that this district is in proximity to the holy shrine of Masoumeh and includes hotels that receive travelers who usually prefer a long stay. District 8 is located a little far from Qom city; in other words, district 8 is a small town in itself having all the necessary facilities and necessities, so the people of this district have little communication with the city center of Qom. Meanwhile, districts 2, 3, 5 and 6 are considered the less prosperous districts.

Furthermore, the results revealed a significant difference in people’s behavior before and after the outbreak of COVID-19, showing the impact of education and emphasis on adhering to health protocols on the part of people. Lack of awareness about disinfectants, for instance the percentage of chlorine in bleaches, affects people’s performance and hence their engagement with this disease. In situations where certain disinfectants are not available, e.g., 70% alcohol, if people are aware of the chlorine content of bleaches, they can use it as a disinfectant to prevent infection with this virus.

In a similar study conducted among COVID-19 patients in Ethiopia, patients’ awareness was found to be in a good range, while the majority of them would obtain information through television and radio. Furthermore, living in rural districts and lack of education [33] were found to constitute one of the variables of weak awareness, which emphasizes public education in villages, deprived districts, and illiterate people. It was also found that those with poor awareness about the disease were 8.6 times more likely to have poor performance [33]. A study in Italy also showed good awareness among students. The study maintains that the lockdown has given people an opportunity to watch television and surf the internet to improve their awareness [32].

The current study showed a significant relationship between citizens’ awareness and performance as well as the level of education, but in another study in Pakistan, the level of education differed from the awareness of participants [30]. Another study in Vietnam also found a link between occupation and awareness in which pharmacists had the highest level of awareness about this disease [34]. The results of the current study are in contrast with those of other research conducted on low-income families in the Philippines in which the level of awareness was good, but the proportion of people who undertook preventive action was low [35]. Furthermore, a study measuring nurses’ awareness reported that the participants’ awareness was at a good level [36]. Nurses and health care providers are among those groups whose information and knowledge should be high enough to prevent the spread of the disease and to educate other people as well. All in all, these results signify that in addition to trying to produce vaccine and treatment, educating and increasing awareness and performance can play a significant role in controlling and preventing this pandemic.

In the current study, municipality district 8 had the lowest incidence rate of the disease while municipality district 7—with a significant difference compared with other municipality districts—had the highest incidence rate as well as lower mean scores in both awareness and performance. Therefore, it might be concluded that increasing awareness and performance does not reduce the incidence, but a decrease in awareness and performance would certainly lead to an increase in the incidence of the disease.

According to some of the above-mentioned studies [29,30,32,33,34,35,36], the awareness and performance levels of most participants were satisfactory to some extent. Therefore, training to control and prevent the spread of this virus must move in a direction that it could not only maintain this process but also increase awareness and performance. As such, there seems to be a need to conduct more research with these goals in mind to better identify the role of awareness and performance.

### Limitations, Strengths and Suggestions for Future Research

Similar to many studies, the current study is not without limitations. To begin with, the research questions in this study were not comprehensive and diverse, so future studies should better include questions of broader and more diverse types. On the other hand, due to the pandemic and the online completion of the questionnaires, the current study was inadvertently biased in which the majority of participants were women, married, in the age range of 31–40, had the education level of a high school diploma and a 1–2 million Toman monthly income. In addition, due to the use of cyberspace, the age range of most of the participants was between groups 31–40 and 20–30 years, which can be explained by the fact that people in these age groups use cyberspace more frequently than other ones. Furthermore, people with education level of high school are among the people who have more presence in cyberspace, which also caused some bias in this study. As such, it is recommended that future studies go in such a direction so as to prevent the occurrence of these biases. Novelty of the subject is the foremost strength of the current study. To date, no study has been conducted to survey awareness and performance regarding use of disinfectants in Iran, esp. the city of Qom.

## 5. Conclusions

The current study found that citizens’ awareness was lower than their performance, indicating the necessity of more training programs in public places, schools, universities, and governmental offices as well as maintaining training about the proper and timely methods of using disinfectants. This finding might be attributed to the fact that people’s behavior during the crisis differed from what was customary in the past prior to the recent outbreak. It was also found that municipality districts with a higher level of education, culture, and income (municipality districts 1 and 4) had higher performance and awareness, compared to others, while the incidence rate was not low. On the other hand, municipality district 7, which mostly contains immigrants and travelers, had the lowest levels of awareness and performance as well as having highest incidence rate with a significant difference compared to other municipality districts. As such, it is concluded that a decrease in incidence is not only achieved by increasing awareness and performance, but an increase in incidence is definitely a result of reduced awareness and performance. Since other factors such as individual nutrition, environment, immunity level, and observance of all health protocols all contribute to incidence rate; and as the research questions were limited to the use of disinfectants in the current study, it is recommended that future studies address the role of above-mentioned factors in addition to the role of awareness and performance in using disinfectants.

## Figures and Tables

**Figure 1 ijerph-18-02099-f001:**
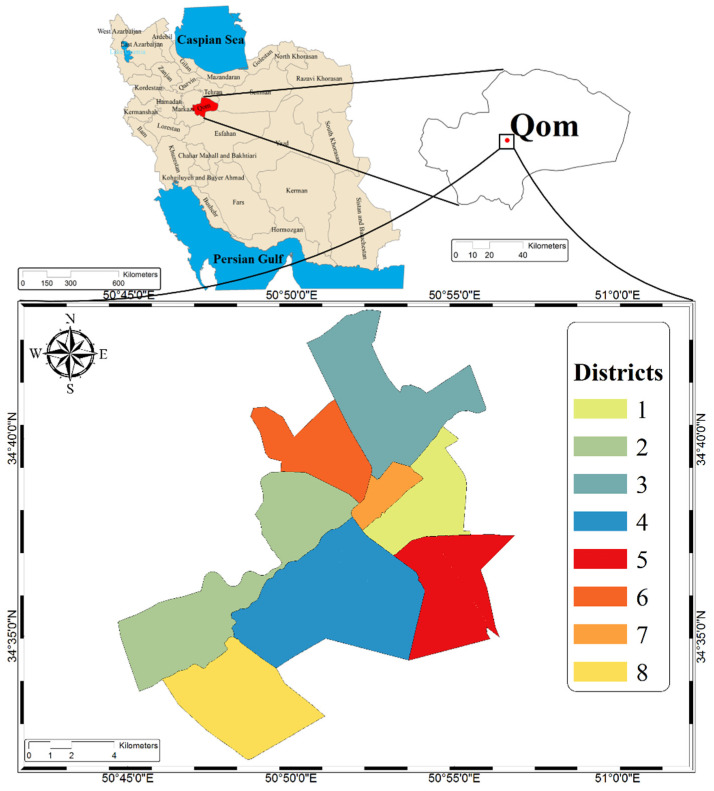
Study area.

**Figure 2 ijerph-18-02099-f002:**
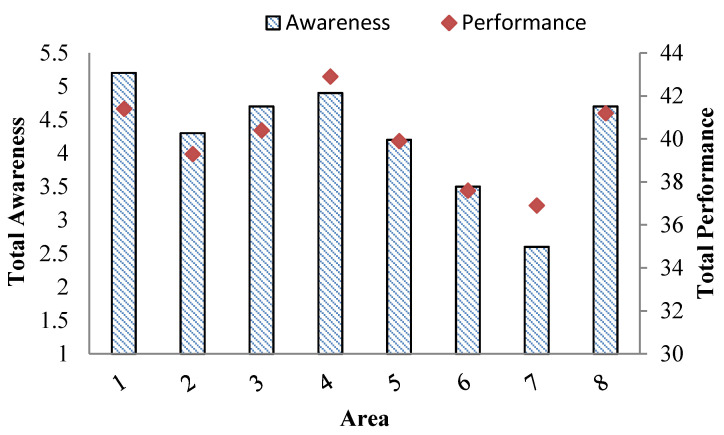
Citizens’ level of awareness and performance.

**Figure 3 ijerph-18-02099-f003:**
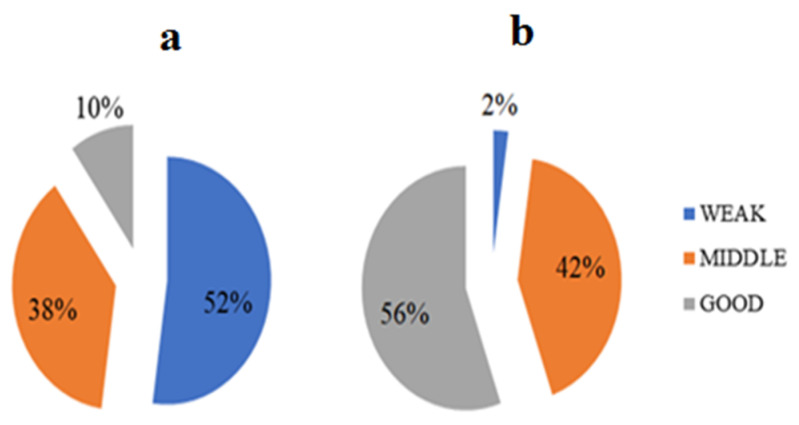
Citizens’ level of awareness (**a**) and performance (**b**).

**Figure 4 ijerph-18-02099-f004:**
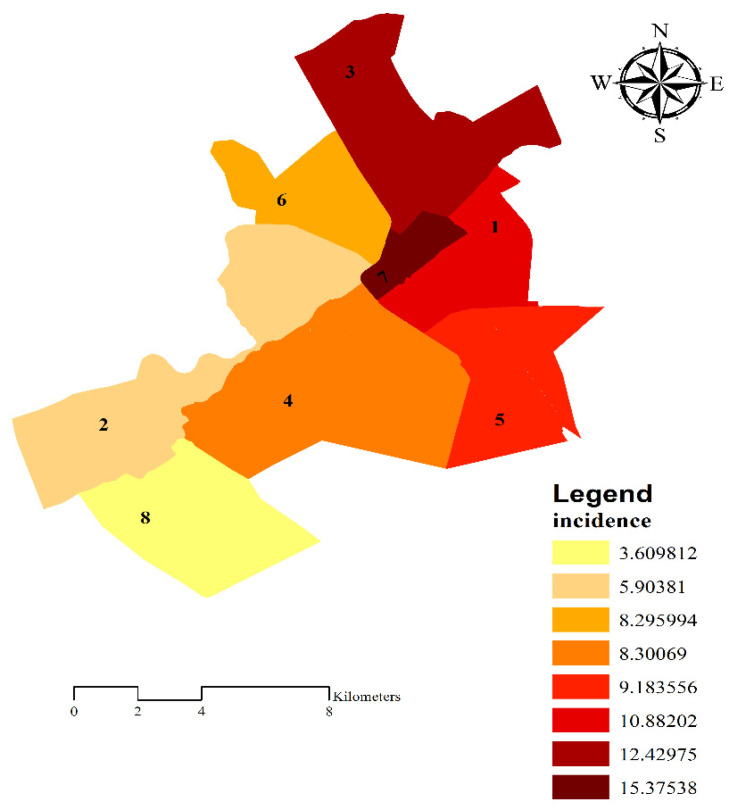
The incidence rate in municipality districts.

**Table 1 ijerph-18-02099-t001:** Socio-demographic information.

Variables	Number	Percent
**Sex**		
Male	128	32
Female	246	61.5
Missing values	26	6.5
**Marital Status**		
Single	51	12.75
Married	333	83.25
Missing values	16	4
**Age (Years)**		
20–30	107	26.75
31–40	171	42.75
41–50	95	23.75
51–60	21	5.25
60+	3	0.75
Missing values	3	0.75
**Municipality Districts**		
1	65	16.25
2	58	14.5
3	48	12
4	75	18.75
5	25	6.25
6	80	20
7	14	3.5
8	35	8.75
Missing values	-	-
**Education Level**		
No education	8	2
Junior High school	74	18.5
High school diploma	143	35.75
Bachelor	127	31.75
Master and Ph.D.	44	11
Missing values	4	1
**Job Status**		
Self-employed	80	20
Governmental employee	79	19.75
Housewife	192	48
Student	9	2.25
Jobless	10	2.5
Missing values	30	7.5
**Income Status**		
Less than 1 m *	115	28.75
1–2 m	120	30
2–5 m	103	25.75
5–10 m	12	3
10+	2	0.5
Missing values	48	12

* Less than 1 million Tomans (Iranian currency).

**Table 2 ijerph-18-02099-t002:** Distribution of responses relating to awareness.

The Questions	Number	Percent
**1. Which alcohol is used as a disinfectant?**		
**Ethanol**	197	49.25
Methanol	37	9.25
Both	44	11
None	39	9.75
Do not know	73	18.25
Missing values	10	2.5
**2. Which one is used for surface disinfection?**		
Sodium hypochlorite	155	38.75
Perchlorine	12	3
Alcohol	73	18.25
**All**	135	33.75
Do not know	21	5.25
Missing values	4	1
**3. Which alcohol is an industrial alcohol which is toxic and deadly?**		
**Methanol**	223	55.75
Ethanol	39	9.75
Both	29	7.25
None	7	1.75
Do not know	96	24
Missing values	6	1.5
**4. How much chlorine is normally present in bleach?**		
100%	23	5.75
20%	36	9
70%	54	13.5
**5%**	61	15.25
Do not know	220	55
Missing values	6	1.5
**5. What is the ratio of bleach to water for making surface disinfection?**		
1 to 5	165	41.25
1 to 2	9	2.25
3 to 1	18	4.5
**1 to 50**	165	41.25
Do not know	39	9.75
Missing values	4	1
**6. For pre-disinfection of fruits and vegetables, how many minutes do they need to be in water and dishwashing liquid?**		
5 to 10	199	49.75
**2 to 5**	118	29.5
60	8	2
30	39	9.75
Do not know	32	8
Missing values	4	1
**7. How many dishwashing liquid drops are needed for pre-disinfection of fruits?**		
**1 to 3**	285	71.25
7 to 10	54	13.5
20 to 30	10	2.5
15 to 20	5	1.25
Do not know	42	10.5
Missing values	4	1
**8. Which one is the most effective concentration of alcohol for disinfection?**		
96%	43	10.75
**70%**	246	61.5
0.50%	8	2
1%	11	2.75
Do not know	86	21.5
Missing values	6	1.5
**9. How long can it take for the disinfectant solution prepared by chlorine to be used for disinfection?**		
1 h	32	8
**24 h**	143	35.75
1 w	33	8.25
1 m	24	6
Do not know	161	40.25
Missing values	7	1.75
**10. At which temperature do you use water to dilute disinfectant solution?**		
Warm	36	9
**Cold**	182	45.5
Tepid	127	31.75
None	8	2
Do not know	43	10.75
Missing values	4	1

The correct answers are in bold type.

**Table 3 ijerph-18-02099-t003:** Total mean scores of awareness and performance by socio-demographic characteristics.

Variables	Total Awareness *			Total Performance ***		
	Mean	SD	*p*-Value **	Mean	SD	*p*-Value **
**Sex**			0.31			0.8
Men	3.8	2.2		39.5	7.6	
Women	4.7	2.1		40.6	7.6	
**Marital Status**			0.06			0.77
Single	4.3	1.9		38	7.5	
Married	4.4	2.2		40.6	7.6	
**Age**			**0.04**			0.1
20–30	4	2.1		38.7	7.5	
31–40	4.7	2		40.4	8.1	
41–50	4.4	2.3		41.1	7.5	
51–60	3.3	2.3		42.6	6.5	
60+	4.3	1.2		43.3	5.6	
**Municipality Districts**			**0.00**			**0.01**
1st	5.1	2.4		41.4	6.6	
2nd	4.3	1.8		39.3	8.3	
3rd	4.7	2.3		40.4	8.6	
4th	4.8	2.2		43	7.1	
5th	4.1	2.1		39.9	7.2	
6th	3.4	1.7		37.6	8.4	
7th	2.6	1.8		36.6	6.9	
8th	4.6	1.7		41.2	6.2	
** Education Level **			**0.00**			**0.01**
No education	3.2	2.2		30.4	8.7	
Junior high school	3.7	2		38.6	8.2	
High school diploma	4.1	2		40.4	7.6	
Bachelor	4.7	2.3		41.1	7.7	
Master and Ph.D.	5.34	2.1		41.5	6.1	
** Job Status **			**0.01**			**0.01**
Self-Employed	3.8	2.1		38.1	8.5	
Governmental-employee	4.8	2.1		41.4	6.9	
Housewife	4.6	2		40.7	7.7	
Student	5	2.6		41.5	4.9	
Jobless	3.9	2.3		35.3	8.2	
** Income Status **						
Less Than 1 m	4.4	2.2	0.53	39.9	8.1	0.33
1–2 m	4.1	2.1		39	8.5	
2–5 m	4.5	2.3		41.1	6.2	
5–10 m	4.5	1.7		42	8.9	
Above10	3	2.8		41	4.2	
**Total**	4.4	2.2		40.2	7.8	

* Awareness is calculated from 10 points. ** Bold *p* values mean a significant difference at level of less than 0.05. *** Performance is calculated from 57 points.

**Table 4 ijerph-18-02099-t004:** Incidence rate in every municipality district.

Municipality District	Population	Patient	Incidence
1	192,060	209	10.9
2	189,708	112	5.9
3	171,363	213	12.4
4	192,755	160	8.3
5	78,401	72	9.2
6	213,356	177	8.3
7	41,625	64	15.3
8	121,890	44	3.6

**Table 5 ijerph-18-02099-t005:** Results of ANOVA (LSD) about the differences between group means of district 7 with other municipality districts in terms of awareness and performance.

Municipality		Standard Error	*p*-Value	95% Confidence Interval	
				Lower Bound	Upper Bound
7	1	0.64	0.00	−3.86	−1.34
	2	0.65	0.01	−2.98	−0.43
	3	0.66	0.001	−3.44	−0.84
	4	0.63	0.00	−3.54	−1.05
	5	0.73	0.03	−3.02	−0.16
	6	0.63	0.05	−2.13	0.35
	8	0.69	0.00	−3.44	−0.74

## Data Availability

Data sharing not applicable.

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
