# Peer review of "Awareness and Performance towards Proper Use of Disinfectants to Prevent COVID-19: The Case of Iran"

_ijerph, 2021, doi:10.3390/ijerph18042099_

Round 1
Reviewer 1 Report
Manuscript reviewer's report:
"Awareness and Performance towards Proper Use of
Disinfectants and its Relationship with Geographical
Distribution of COVID-19 Incidence"
The study reflects the results of a survey on the use of disinfectants to protect and prevent coronavirus infection.
Page 1 line 1
"Awareness and Performance towards Proper Use of
Disinfectants and its Relationship with Geographical
Distribution of COVID-19 Incidence"
In my opinion, the title is too detailed.
(Covid-19 prevention awareness and performance: the case of Iran)
p.12 l.221
The use of the following term is also appropriate: "cases per 10000 population"
abstract
Throughout the manuscript
p.1 l.30
p.1 l.95
region vs area vs disctrict
Please, consistently apply one of these terms
p.2 l.141
"89 of which were excluded due to the lack of necessary information and distortion of information (including those respondents under the age
of 20). "
Why are respondents under the age of 20 excluded?
p.4 l.121-123
" This section also included questions about people's health
performance regarding proper and timely use of antiseptic agents before COVID-19 crisis, like “Did
you use to do it?” followed by “yes” and “no” options...."
And the question in questionnaire number 34
"Did you use to apply special disinfection solutions to disinfect fruit and vegetables?"
The wording of this question does not explicitly refer to the time period "before the COVID-19 pandemic". It is unlikely that the answers can be interpreted to compare "before the pandemic-after pandemic".
p.7 193 table 3
In my opinion, this is a very long table, longer than 2 pages, it can be placed in the app.
p.13 table5 4st st? Please check.
5st st? Please check.
6st st? Please check.
7st st? Please check.
8st st? Please check.
13 l.255
"but their performance was similar to the that of the participants"
3.3. performance
Table 6: results
Please, check
Author Response
Date: Wednesday, January 6, 2021
Dear Editor
Enclosed is a paper, entitled “Awareness and Performance towards Proper Use of Disinfectants to Prevent COVID-19: the case of Iran”. This study investigated the awareness and performance of Qom citizens about using disinfectants and compared their relationship with geographical distribution of corona outbreak in Qom, Iran. The highest incidence rate of COVID-19 was in region 7, which had the lowest mean score in both awareness and performance. The results showed that region 7 had a significant difference compared with other regions, with lower mean scores in awareness and performance.
Given the novelty of this research in Qom, we believe this paper would make an important contribution to science. Hence, it can be regarded as appropriate for publication in the International Journal of Environmental Research and Public Health. Finally, this paper is our original, unpublished work and has not been submitted to any other journal for review.
Dear reviewer 1
We appreciate the effort you made to improve the manuscript and thank you for your comments and corrections. A minor revision of the paper has been carried out to take all of them into account. We believe the paper has been significantly improved.
According to the journal guidelines, we detail the minor changes (highlighted in yellow) that have been made in the paper in order to ameliorate the minor weaknesses identified by the review process.
We hope all comments have been addressed favorably. It should be mentioned that the comments and replies have been presented in detail in the table below.
We also thank the reviewer 2 for accepting our manuscript.
Sincerely,
Corresponding authors
|
Number |
Comments And Suggestions |
Replies |
|
1 |
Page 1 line 1 "Awareness and Performance towards Proper Use of |
Thanks for your comment. The title was changed regarding your suggestion. “Awareness and Performance towards Proper Use of Disinfectants to Prevent Covid-19 : the case of Iran” |
|
2 |
p.12 l.221 |
Thanks for your comment. Regarding your comment, the sentence was changed to the following sentence: results showed that the highest rate of incidence of COVID-19 was in municipality district 7 (15.4 cases per 10000 population), while the lowest was in municipality district 8 (3.6 cases per 10000 population). You can find it in line 225. |
|
3 |
abstract |
Thanks for your comment. Regarding your comment, all synonymous words were replaced with district. |
|
4 |
p.2 l.141 |
Thanks for your comment. In the messages that we sent to individuals and groups in cyberspace with the questionnaire, we emphasized that only people over the age of 20 should answer the questions, but nevertheless, a number of questionnaires were filled by those under the age of 20. The reason is that people over 20 answer questions more seriously, so the answers are more credible. Also, the behavior of people over 20 is more effective in the family. |
|
5 |
p.4 l.121-123 And the question in questionnaire number 34 |
Thanks for your comment. The questions were changed to “Did you use to do it (before the COVID-19 pandemic)?” In our language, Persian, the questions were clear and represented the concept of “before the pandemic”. |
|
6 |
p.7 193 table 3 |
Thanks for your comment. Regarding your comment, table 3 was placed in app as table S1. |
|
7 |
p.13 table5 4st st? Please check. |
Thanks for your comment. We mean the municipality districts. The “st” was removed. |
|
8 |
13 l.255 |
Thanks for your comment. The sentence was changed to the following sentence: In a study conducted by Taghrir.et al on medical students, it was found that a great deal of participants’ awareness and performance was at a high level, which may be due to their job and profession. You can find it in line 262. |
|
9 |
3.3. performance |
(Due to removing table 3, the table 6 was changed to table 5), .Table 5 shows the results of ANOVA (LSD) about the differences between group means of district 7 with other municipality districts in terms of awareness and performance. This is not related to 3.3.performance. |
Authors
- First Name: Zahra
Surname: Safari
Affiliation: 1. Research Center for Environmental Pollutants and Department of Environmental Health Engineering, Qom University of Medical Sciences, Qom, Iran.
- Student Research Committee, Qom University of Medical Sciences, Qom, Iran
E-mail: zsafari353@gmail.com, zsafari@muq.ac.ir
- First Name: Reza
Surname: Fouladi-Fard
Affiliation: Research Center for Environmental Pollutants and Department of Environmental Health Engineering, Qom University of Medical Sciences, Qom, Iran.
E-mail: rfouladi@muq.ac.ir, rezafd@yahoo.com.
- First Name: Razieh
Surname: Vahidmoghadam
Affiliation: Research Center for Environmental Pollutants and Department of Environmental Health Engineering, Qom University of Medical Sciences, Qom, Iran.
E-mail: r1375vahid@gmail.com
- First Name: Mohammad Reza
Surname: Hosseini
Affiliation: Student Research Committee, Qom University of Medical Sciences, Qom, Iran
E-mail: m.reza68.h@gmail.com
- First Name: Abolfazl
Surname: Mohammadbeigi
Affiliation: Department of Public Health, Faculty of Health, Qom University of Medical Sciences, Qom, Iran
E-mail: beigi60@gmail.com
- First Name: Alireza
Surname: Omidi Oskouei
Affiliation: Department of Public Health, Faculty of Health, Qom University of Medical Sciences, Qom, Iran
E-mail: omidioskouei@gmail.com
- First Name: Mostafa
Surname: Rezaali
Affiliation: Independent researcher, Isfahan, Iran (formerly: Department of Civil and Environmental Engineering, Qom, Iran)
E-mail: mostafarezaali@gmail.com
- First Name: Ferrante
Surname: Margherita
Affiliation: Department of Medical, Surgical and Advanced Technologies "G.F. Ingrassia", University of Catania, Catania, Italy. Via Santa Sofia, 87 – 95123 Catania.
E-mail: marfer@unict.it
- First Name: Maria
Surname: Fiore
Affiliation: Department of Medical, Surgical and Advanced Technologies "G.F. Ingrassia", University of Catania, Catania, Italy. Via Santa Sofia, 87 – 95123 Catania.
E-mail: mfiore@unict.it
Corresponding Authors:
Reza Fouladi-Fard
Address: Department of Environmental Health Engineering, School of Health, Qom University of Medical Sciences.
Telefax: +982537833361
Margherita Ferrante
Affiliation: Department of Medical, Surgical and Advanced Technologies "G.F. Ingrassia", University of Catania, Catania, Italy. Via Santa Sofia, 87 – 95123 Catania

Reviewer 2 Report
The manuscript “Awareness and Performance towards Proper Use of Disinfectants and its Relationship with Geographical Distribution of COVID-19 Incidence” focuses on a very important subject related to the greatest scourge of our times and also, consequently, the greatest object of investigation a world level: COVID-19.
Since COVID-19 is a highly contagious disease, for which no treatment is known, and it is only now possible to initiate a large-scale vaccination, the available defense for populations is to reduce contagion rates as much as possible, by adopting proper hygiene, disinfection, and social behavior.
It is in this environment that the manuscript under analysis is located, in which the authors analyze awareness and performance in relation to the proper use of disinfectants and their relationship with the geographic distribution of the incidence of COVID-19, in an Iranian city.
The importance of this type of studies is evident, as they can serve to detect the level of performance of the populations, and, consequently, inform them with a view to adopting and improving the procedures.
Therefore, it is important that these studies are scientifically rigorous, and that their results are presented in a simple way, so that they can be understood by a large number of people.
Thus, after an Introduction in which the authors are very well informed on the subject, they present the problem clearly, and describe the study they intend to carry out, drawing relevant bibliography and similar studies, going to the Material and Methods. In this, having taken care of ethical problems, they describe the procedures they will follow. It is about collecting data through a questionnaire, in order to make its statistical analysis.
The questionnaire is well structured and appears to be written in a simple and understandable way for the respondents. The quantitative techniques used are simple and appropriate to the problem. Its simplicity contributes to your better understanding.
The results are well presented, in a perfectly intelligible way. Their discussion is fruitful and clarifying and the conclusions clear and appropriate.
Bibliography is necessary and sufficient.
Written in simple and intelligible English, the manuscript deserves to be published as is.
Author Response
Date: Wednesday, January 6, 2021
Dear Editor
Enclosed is a paper, entitled “Awareness and Performance towards Proper Use of Disinfectants to Prevent COVID-19: the case of Iran”. This study investigated the awareness and performance of Qom citizens about using disinfectants and compared their relationship with geographical distribution of corona outbreak in Qom, Iran. The highest incidence rate of COVID-19 was in region 7, which had the lowest mean score in both awareness and performance. The results showed that region 7 had a significant difference compared with other regions, with lower mean scores in awareness and performance.
Given the novelty of this research in Qom, we believe this paper would make an important contribution to science. Hence, it can be regarded as appropriate for publication in the International Journal of Environmental Research and Public Health. Finally, this paper is our original, unpublished work and has not been submitted to any other journal for review.
We also thank the reviewer 2 for accepting our manuscript.
Reviewer 3 Report
The manuscript entitled "Awareness and Performance towards Proper Use of Disinfectants and its Relationship with Geographical Distribution of COVID-19 Incidence" contains new and significant information regarding a very current topic. The originality of the study and the novelty it brings in the field is of actuality. The purpose of the article and its significance is stated clearly. This article aims to draw the attention of all the pawns involved in the public health, regarding the safety procedures that need to be adopted for citizens' awareness. The pandemic has put to a test our society, thus the strategies for maintain the security of population need to be explore.
All the tables are necessary, presented in the appropriate way and can help the reader follow the flow of the manuscript.The results are presented clearly and analysed appropriately.
However, there were some minor issues that need to be addressed before publication. Therefore I recommend the manuscript to be published after minor revision.
- I recommend to introduce some information regarding SARS-CoV-2 health implications. In this regard, I kindly recommend the next paper to be consulting for the introduction section: https://www.mdpi.com/2071-1050/13/1/150
- The population taken in the study is representative for all districts? Also, since women constituted the majority of sample, ths fact could have a significant influence on the study? I believe that authors should state the limitations of this study.
Author Response
Date: Wednesday, January 6, 2021
Dear Editor
Enclosed is a paper, entitled “Awareness and Performance towards Proper Use of Disinfectants to Prevent COVID-19: the case of Iran”. This study investigated the awareness and performance of Qom citizens about using disinfectants and compared their relationship with geographical distribution of corona outbreak in Qom, Iran. The highest incidence rate of COVID-19 was in region 7, which had the lowest mean score in both awareness and performance. The results showed that region 7 had a significant difference compared with other regions, with lower mean scores in awareness and performance.
Given the novelty of this research in Qom, we believe this paper will be an important contribution to science. Hence, it can be regarded as appropriate for publication in the International Journal of Environmental Research and Public Health. Finally, this paper is our original, unpublished work and has not been submitted to any other journal for review.
Dear reviewer 3
We appreciate the effort you made to improve the manuscript and thank you for your comments and suggestions. A minor revision of the paper has been carried out to take all of them into account. We believe the paper has been significantly improved.
According to the journal guidelines, we detail the minor changes (highlighted in green) that have been made in the paper in order to ameliorate the minor weaknesses identified by the review process.
We hope all comments have been addressed favorably. It should be mentioned that comments and replies have been presented in detail in the table below.
Sincerely,
Corresponding authors
|
Number |
Comments And Suggestions |
Replies |
|
1 |
I recommend to introduce some information regarding SARS-CoV-2 health implications. In this regard, I kindly recommend the next paper to be consulting for the introduction section: https://www.mdpi.com/2071-1050/13/1/150 |
Thanks for your comment. Your suggestion has been applied in line 47. |
|
2 |
1. The population taken in the study is representative for all districts? 2. Also, since women constituted the majority of sample, this fact could have a significant influence on the study? I believe that authors should state the limitations of this study. |
1. Thanks for your comment. The number of questionnaires in each district was filled according to population of that district. Therefore, the population that entered the study in each district, is representative of that district. We covered all municipality districts of Qom so that this population is representative of Qom city. 2. Thanks for your comment. We did not impose any limitations other than those participants over the age of 20. However, due to the fact that most of the questionnaires were filled online, there were more female participants than men, so our study was inadvertently biased and women became the majority of participants. The same explanation was added in the discussion section. You can find it in line 243. |
Authors
- First Name: Zahra
Surname: Safari
Affiliation: 1. Research Center for Environmental Pollutants and Department of Environmental Health Engineering, Qom University of Medical Sciences, Qom, Iran.
- Student Research Committee, Qom University of Medical Sciences, Qom, Iran
E-mail: zsafari353@gmail.com, zsafari@muq.ac.ir
- First Name: Reza
Surname: Fouladi-Fard
Affiliation: Research Center for Environmental Pollutants and Department of Environmental Health Engineering, Qom University of Medical Sciences, Qom, Iran.
E-mail: rfouladi@muq.ac.ir, rezafd@yahoo.com.
- First Name: Razieh
Surname: Vahidmoghadam
Affiliation: Research Center for Environmental Pollutants and Department of Environmental Health Engineering, Qom University of Medical Sciences, Qom, Iran.
E-mail: r1375vahid@gmail.com
- First Name: Mohammad Reza
Surname: Hosseini
Affiliation: Student Research Committee, Qom University of Medical Sciences, Qom, Iran
E-mail: m.reza68.h@gmail.com
- First Name: Abolfazl
Surname: Mohammadbeigi
Affiliation: Department of Public Health, Faculty of Health, Qom University of Medical Sciences, Qom, Iran
E-mail: beigi60@gmail.com
- First Name: Alireza
Surname: Omidi Oskouei
Affiliation: Department of Public Health, Faculty of Health, Qom University of Medical Sciences, Qom, Iran
E-mail: omidioskouei@gmail.com
- First Name: Mostafa
Surname: Rezaali
Affiliation: Independent researcher, Isfahan, Iran (formerly: Department of Civil and Environmental Engineering, Qom, Iran)
E-mail: mostafarezaali@gmail.com
- First Name: Ferrante
Surname: Margherita
Affiliation: Department of Medical, Surgical and Advanced Technologies "G.F. Ingrassia", University of Catania, Catania, Italy. Via Santa Sofia, 87 – 95123 Catania.
E-mail: marfer@unict.it
- First Name: Maria
Surname: Fiore
Affiliation: Department of Medical, Surgical and Advanced Technologies "G.F. Ingrassia", University of Catania, Catania, Italy. Via Santa Sofia, 87 – 95123 Catania.
E-mail: mfiore@unict.it
Corresponding Authors:
Reza Fouladi-Fard
Address: Department of Environmental Health Engineering, School of Health, Qom University of Medical Sciences.
Telefax: +982537833361
Margherita Ferrante
Affiliation: Department of Medical, Surgical and Advanced Technologies "G.F. Ingrassia", University of Catania, Catania, Italy. Via Santa Sofia, 87 – 95123 Catania

This manuscript is a resubmission of an earlier submission. The following is a list of the peer review reports and author responses from that submission.
Round 1
Reviewer 1 Report
- lines 1/2: Because the risk of death is higher for people with underlying diseases, people with respiratory failure need basic care [6]
- why?; the second part part of the statement does not follow from the first
- please fix
- line 10: “This is especially crucial for situations in which the patient is quarantined at home [10]”
- why?
- according to [10] this is important for people taking care of quarantined people
- please fix
- lines 63/64:Thorough hand washing is one of the most significant measures to prevent transmission of infection, which must be done carefully
- this does not reflect the current knowledge and CDC recommendations
- please update
- line 79/80: “Due to the pandemic of coronavirus and the need to pay attention to health care (including proper and timely use of disinfectants), which according to recommendations of WHO, is one of the basic principles of virus prevention and control”
- confusing sentence
- please fix
- lines 100/101 state “it was decided that the questionnaires be designed and completed online.”
- however, lines 122/123 then state “Due to the completion of the appropriate number of questionnaires according to the population of each region, in addition to the use of cyberspace, several questionnaires were completed in person.”
- this should be addressed in a single paragraph and should be better explained
- also, how did in-person interviews affect responses?
- please attach the actual questionnaire in appendix
- otherwise the reader has no idea what this really refers to“Questions 36 to 40 dealt with whether people receive health information”
- table 3: please explain how you collapsed the 5 categories into 2 to compare them with yes/no?
- Figure 2: red line is misleading
- please redo figure without this line
- line 207: most important finding does not have its own table and/or figure and is not properly analyzed.
- please elaborate and provide either a table or figure
- Table 5: how are these districts different in terms of demographics?
- this information is crucial to understanding the results
- please provide this information
- lines 222-224: is the incidence different for women than for men?
- is the anything known about infection networks?
- please explain your findings and address the above questions
- line 228 “citizens' awareness was lower than their performance”
- please elaborate on the implication of this finding?
- lines 234-249 … very confusing and at times seemingly contradictory
- lines 238/239 "According to the results, there was a strong correlation (with rho 0.95) between the total mean of awareness and performance" contradict line 234
- please clarify
- lines 238/239 "According to the results, there was a strong correlation (with rho 0.95) between the total mean of awareness and performance" contradict line 234
- line 257: “and surf the internet to improve their awareness” …
- this is mentioned as a(n implicit) recommendation but ignores the need for internet access
- given the differences between districts, how successful would such a strategy be? please explain
- earlier in the manuscript you mention that television is the main source of information? how is this strategy compatible with that finding? please explain
- this is mentioned as a(n implicit) recommendation but ignores the need for internet access
- lines 219 - 269: How are the populations mentioned in this section different/similar
- right now these are just random comparisons
- this sections needs a lot of work to present a coherent argument
Author Response
Response to Reviewer 1 Comments
Dear Reviewer
Thank you for your comments, corrections and suggestions. We highly appreciate the effort you made in order to improve this manuscript.
A major revision of the paper has been carried out to take all the points into account. We believe the paper has been significantly improved.
According to the journal guidelines, we detail the major changes (highlighted in green) that have been made in the paper to correct the main weaknesses identified by the review process.
We hope all comments are addressed favorably.
Sincerely,
Corresponding authors
Point 1: Lines 1/2: Because the risk of death is higher for people with underlying diseases, people with respiratory failure need basic care [6] why? The second part of the statement does not follow from the first. Please fix
Response 1: Thanks for your comment. According to the reviewer, it was replaced with the following sentence: The risk of respiratory failure requiring exigent care support in patients infected with SARS-CoV-2 is crucial.
Point 2: Line 10: “This is especially crucial for situations in which the patient is quarantined at home [10]” Why? according to [10] this is important for people taking care of quarantined people .please fix
Response 2: Thanks for your comment, the previous sentence was misunderstood, so it was replaced with the following sentence: This is especially crucial for people who take care of quarantined patients at home.
Point 3: Lines 63/64: Thorough hand washing is one of the most significant measures to prevent transmission of infection, which must be done carefully. This does not reflect the current knowledge and CDC recommendations. Please update
Response 3: Thanks for your comment, since the subject of our study is disinfectants and these days, the most important strategy is using mask. We corrected the sentence according to the subject of study: Thorough hand washing is the usual measure to prevent transmission of infection, which must be done carefully.
Point 4: Line 79/80: “Due to the pandemic of coronavirus and the need to pay attention to health care (including proper and timely use of disinfectants), which according to recommendations of WHO, is one of the basic principles of virus prevention and control” confusing sentence. please fix
Response 4: Thanks for your comment, it was changed: According to recommendations of WHO, health care (including proper and timely use of disinfectants) is one of the basic principles of virus prevention and control due to the need to pay attention to health care during pandemic of COVID-19.
Point 5: Lines 100/101 state “it was decided that the questionnaires be designed and completed online.” However, lines 122/123 then state “Due to the completion of the appropriate number of questionnaires according to the population of each region, in addition to the use of cyberspace, several questionnaires were completed in person.” This should be addressed in a single paragraph and should be better explained. Also, how did in-person interviews affect responses?
Response 5: Thanks for your comment, in-person interviews do not affect responses, we do not mean this way affects responses, we printed them just due to the completion of the appropriate number of questionnaires according to the population of each municipality district. The sentences were separated in a paragraph with more detailed explanation.
Point 6: Please attach the actual questionnaire in appendix
Response 6: Thanks for your comment. The questionnaire proposed revision has been attached in Supplementary material (S1).
Point 7: Table 3: please explain how you collapsed the 5 categories into 2 to compare them with yes/no?
Response 7: Thanks for your comment. We changed the table for better comprehension. We randomly asked if they had done this habit before the crisis, and any question answered with yes or no, is only related to the previous question.
Point 8: Figure 2: red line is misleading. Please redo figure without this line.
Response 8: Thanks for your comment. The proposed revision has been applied.
Point 9: Line 207: most important finding does not have its own table and/or figure and is not properly analyzed. Please elaborate and provide either a table or figure.
Response 9: Thanks for your comment. The proposed revision has been applied.
Point 10: Table 5: how are these districts different in terms of demographics? This information is crucial to understanding the results. Please provide this information.
Response 10: Thanks for your comment. The previous figure1, 4 was misunderstood due to division of areas in each municipality district. So, we removed division of areas and population distribution is according to Table 5.
Point 11: Lines 222-224: is the incidence different for women than for men? Is the anything known about infection networks? Please explain your findings and address the above questions.
Response 11: Thanks for your comment. Due to the correspondence with the health center, they did not allow to access to patients in detail. We only had access on total number of patients in each municipality district.
Point 12: Line 228 “citizens' awareness was lower than their performance” please elaborate on the implication of this finding?
Response 12: Thanks for your comment. The proposed revision has been applied.
Point 13: Lines 234-249 … very confusing and at times seemingly contradictory. Lines 238/239 "According to the results, there was a strong correlation (with rho 0.95) between the total mean of awareness and performance" contradict line 234. Please clarify.
Response 13: Thanks for your comment. According to the results, there was a strong correlation (with rho 0.95) between the total mean of awareness and the total mean of performance in each municipality district, and those districts with more awareness performed better.
Point 14: Line 257: “and surf the internet to improve their awareness” … this is mentioned as a (n implicit) recommendation but ignores the need for internet access. Given the differences between districts, how successful would such a strategy be? Please explain. Earlier in the manuscript you mention that television is the main source of information? How is this strategy compatible with that finding? Please explain.
Response 14: Thanks for your comment. The sentence: (the lockdown has given people an opportunity to watch television and surf the internet to improve their awareness) is not our results. It is directly referenced to [26]. Qom regions, as mentioned in discussion, are different in terms of demographics, culture, education, income level. But all regions have access to the Internet and most people have access to the Internet as well. The fact that our results showed television has played a greater role in educating people, shows more trust in mass media. This fact does not reflect that people do not use the Internet at all. As shown in the results, Telegram had the highest percentage among applications after television.
Point 15: Lines 219 - 269: How are the populations mentioned in this section different/similar? Right now these are just random comparisons. This sections needs a lot of work to present a coherent argument.
Response 15: Thanks for your comment. Considering the problems in Figures 1 and 4, by correcting them and the explanations given in response to Comment 9, we hope these problems have also been resolved.

Reviewer 2 Report
This study is based on an on-line questionnaire to evaluate the awareness and performance of Qom citizens about using disinfectants and compare their relationship with geographical distribution of corona outbreak in Qom, Iran. The survey was embedded within sent messages and the citizens were informed of the purposes of the study.
The topic and the purposes of the study are current and important. The introduction provides sufficient background. The questionnaire is more of an intelligence test, do not reflect the everyday practice in terms of disinfection habits. People buy the disinfectant and read the instructions for use.
Some of the questions are misinterpreted:
- How many dishwashing liquid drops are needed for pre-disinfection of fruits? 1to3 7to10 20to30 15to20 Do not know
- Which one is the most effective concentration of alcohol for disinfection? 96% 70% 0.5% 1% Do not know
The parts of the statistical methods, results and discussion are appropriate
English language usage is poor.
There are many grammatical and syntax errors throughout the manuscript.
There are confusing usage of Covid-19 and coronavirus and SARS-CoV-2. There is supposed to be a consistent usage of definitions.
Line 44 the SARS-CoV-2 is the Covid-19 not a group of viruses.
Line 44 the word should be confirmed not "approved".
Line 45, patients need complex care not "basic care".
Line 66, the sequence of words are confusing. Hospitals are using chemicals for disinfectants.
Line 69 the word is supposed to be consistently not "constantly".
Line 73, the expression therefore or in the first place are exclusive for one another.
Line 77 to 82 should be five sentences not two.
This is just the Introduction.
The poor usage of the English language compromises the understanding of the paper.
The questionnaire should be in the appendix in its entirety not a quote by paragraph.
Author Response
Response to Reviewer 2 Comments
Dear Reviewer
Thank you for your comments, corrections and suggestions. We appreciate the effort you made in order to improve this manuscript.
A major revision of the paper has been carried out to take all of them into account. We believe the paper has been significantly improved.
According to the journal guidelines, we detail the major changes (highlighted in blue) that have been made in the paper to correct the main weaknesses identified by the review process.
We hope all comments are addressed favorably.
Sincerely,
Corresponding authors
Point 1: This study is based on an on-line questionnaire to evaluate the awareness and performance of Qom citizens about using disinfectants and compare their relationship with geographical distribution of corona outbreak in Qom, Iran. The survey was embedded within sent messages and the citizens were informed of the purposes of the study.
Response 1: Thanks for your comment.
Point 2: The topic and the purposes of the study are current and important. The introduction provides sufficient background. The questionnaire is more of an intelligence test, do not reflect the everyday practice in terms of disinfection habits. People buy the disinfectant and read the instructions for use.
Response 2: Thanks for your comment. This study was conducted when Iran was facing a shortage of disinfecants and people would make their own hand, surface and clothes sanitizers in the house (from bleach and Ethanol alcohol), based on information they would get from television or the Internet. Therefore, there could be a mistake in making them, using them and the storage time. So, the questions were designed for this purpose.
Point 3: Some of the questions are misinterpreted:
- How many dishwashing liquid drops are needed for pre-disinfection of fruits? 1to3 7to10 20to30 15to20 Do not know
- Which one is the most effective concentration of alcohol for disinfection? 96% 70% 0.5% 1% Do not know
Response 3: Thanks for your comment. As mentioned above, this study was conducted when Iran was facing a shortage of disinfections and people would make their own hand, surface and clothes sanitizers in the house (from bleach and Ethanol alcohol), based on information they would get from television or the Internet. Therefore, there could be a mistake in making them, using them and storage time. So, the questions were designed for this purpose.
Point 4: The parts of the statistical methods, results and discussion are appropriate.
Response 4: Thanks for your comment.
Point 5: English language usage is poor. There are many grammatical and syntax errors throughout the manuscript.
Response 5: Thanks for your comment. The proposed corrections have been applied (English structural corrections and revisions were highlighted in yellow color).
Point 6: There are confusing usage of Covid-19 and coronavirus and SARS-CoV-2. There is supposed to be a consistent usage of definitions.
Response 6: Thanks for your comment. The proposed corrections have been applied.
Point 7: Line 44 the SARS-CoV-2 is the Covid-19 not a group of viruses.
Response 7: Thanks for your comment. The proposed corrections have been applied.
Point 8: Line 44 the word should be confirmed not "approved".
Response 8: Thank for your comment. The proposed corrections haves been applied.
Point 9: Line 45, patients need complex care not "basic care".
Response 9: Thanks for your comment. According to comment of reviewer 1, the sentence has been completely changed.
Point 10: Line 66, the sequence of words are confusing. Hospitals are using chemicals for disinfectants.
Response 10: Thanks for your comment. The proposed corrections have been applied.
Point 11: Line 69 the word is supposed to be consistently not "constantly".
Response 11: Thanks for your comment. The proposed correction has been applied.
Point 12: Line 73, the expression therefore or in the first place are exclusive for one another.
Response 12: Thanks for your comment. The proposed correction has been applied..
Point 13: Line 77 to 82 should be five sentences not two.
Response 13: Thanks for your comment. The proposed correction has been applied.
Point 14: The poor usage of the English language compromises the understanding of the paper.
Response 14: Thanks for your comment. The manuscript was thoroughly revised by an English expert
Point 15: The questionnaire should be in the appendix in its entirety not a quote by paragraph.
Response 15: Thanks for your comment. The proposed revision has been applied.

Round 2
Reviewer 1 Report
line 31 ("It is concluded ...") does not follow from lines 29-31. Please fix
line 38: "SARS-CoV-2 virus (COVID-19)" ... one is a virus, the other the disease that this virus causes ... please be precise and fix
line 39: not "a" but "the" Huanan ... please fix
lines 43/44: what does this mean "In principle, COVID-19 ..." Is it a respiratory disease or not?
line 44: this is a very old paper and improperly cited; 1) please fix citation (and any other incorrect citation); 2) provide more up-to-date data and in particular data specific to Iran
lines 45/46: this sentence is very misleading: "Risk of respiratory failure is crucial"???
line 56: what does this actually mean: "[...] dealing with a virus is a public health approach" you certainly mean the disease caused by a virus
lines 57/58: where are theses surfaces and devices? what kind of devices: this statement is way too oversimplyfying
lines 65/66: this statement does not hold anymore; a lot is known about covid-19 and its ways of transmission
Please explain why you picked Qom. What is special about Qom? Is it representative for other urban areas in Iran?
Reviewer 2 Report
The topic of the manuscript is current and improtant. The English language and style has been corrected well.